# Body Mass Index Distribution in Female Child, Adolescent and Adult Inpatients with Anorexia Nervosa—A Retrospective Chart Review

**DOI:** 10.3390/nu16111732

**Published:** 2024-05-31

**Authors:** Gertraud Gradl-Dietsch, Triinu Peters, Adrian Meule, Johannes Hebebrand, Ulrich Voderholzer

**Affiliations:** 1Department of Child and Adolescent Psychiatry, Psychosomatics and Psychotherapy, University Hospital Essen, University of Duisburg-Essen, Wickenburg Str. 21, 45147 Essen, Germany; triinu.peters@uni-due.de (T.P.); johannes.hebebrand@uni-due.de (J.H.); 2Center for Translational Neuro and Behavioral Sciences, University Hospital Essen, University of Duisburg-Essen, 45147 Essen, Germany; 3Department of Psychology, University of Regensburg, Universitätsstr. 31, 93053 Regensburg, Germany; adrian.meule@med.uni-muenchen.de; 4Department of Psychiatry and Psychotherapy, LMU University Hospital, LMU Munich, 80539 Munich, Germany; uvoderholzer@schoen-klinik.de; 5Schoen Clinic Roseneck, 83209 Chiemsee, Germany

**Keywords:** Anorexia nervosa, BMI distribution, inpatients, adolescents, adults

## Abstract

Background: The variation in body mass index (BMI) of inpatients with anorexia nervosa has not been analyzed across the age span. A positive correlation between BMI and age has been reported in adolescent inpatients aged 15 years and younger that levels off at 15 to 18 years. BMIs standardized for age and sex (standard deviation scores, SDSs) were negatively correlated with age in these inpatients aged 8 to 18 years. Methods: The aims of the current retrospective study were threefold: first, to confirm the relationships of BMI, BMI-SDS and age in adolescent inpatients in a larger sample; second, to systematically assess the relationship of BMI, BMI-SDS, body height-SDS and age in adult inpatients at the time of referral; and third, to assess body height-SDSs and age to evaluate stunting. Results: We included 1001 girls (aged 12–17.9 years) and 1371 women (aged 18–73 years) admitted to inpatient treatment between 2014 and 2021. Mean BMI at admission was 14.95 kg/m^2^ (SD = 1.43; range 10.67–18.47) in adolescents and 14.63 kg/m^2^ (SD = 2.02; range 8.28–18.47) in adults. None of the adolescent patients but 20 adults had very low BMI values below 10 kg/m^2^. Adolescents showed a small but significant positive correlation between age and BMI (r = 0.12; *p* = 2.4 × 10^−4^). In adults, BMI was not correlated with age (r = −0.03; *p* = 0.3). BMI-SDSs was negatively correlated with age in adolescents and less so in adults (r = −0.35; *p* < 0.001 and r = −0.09; *p* = 0.001). Curve fit analyses for all patients indicated that there was a quadratic (age × age) relationship between age and BMI-SDS. Height correlated positively with BMI in adult (r = 0.1; *p* < 0.001) and adolescent (r = 0.09 *p* = 0.005) patients and we detected no evidence for stunting. Conclusions: In conclusion, the BMI of inpatients seems to be relatively stable across the age span with mean values between 14 and 15 kg/m^2^. BMI values initially increase with age in younger patients, drop between ages 18 and 23 and then slowly decline with age.

## 1. Introduction

In a previous study using data from the German Registry for Anorexia Nervosa, we found a clear age dependency of both admission body mass index (BMI) and BMI-standard deviation scores (SDSs) in female inpatients under the age of 19 years [1]. BMI increased to age 15 and subsequently plateaued up to age 18. Only the BMI of patients under 15 years of age showed a moderate correlation with age (r ≈ 0.5). BMI-SDS was negatively correlated with age, implying that older patients had a BMI more deviant from their age group [1]. A possible explanation is the increase in fat mass during puberty by approximately 50% [2], which would allow older adolescents to lose more fat mass.

The question that arises is whether BMI values continue to plateau into adulthood or if further age-dependent changes occur after age 18. Eating disorders have long been considered to primarily affect young people and up until the DSM III a diagnosis of anorexia nervosa (AN) could only be made in people aged under 30 years [3]. Despite the recognition of AN across the lifespan, we are not aware of studies that have systematically assessed the age-dependent BMI distribution of adult patients. Peak age at onset is 15.5 years and the majority of patients develop symptoms before the age of 25 years [4]. With more than 20% of patients continuing to have an eating disorder on long-term follow-up [5], many patients are of adult age.

Key features of AN include restriction of energy intake relative to requirements leading to a significantly low body weight in the context of age, sex, developmental trajectory and physical health [6]. Malnutrition is reported to be associated with the inability to appropriately mineralize and strengthen bone, growth deceleration and stunting [7,8]. Animal models of malnutrition show a delayed epiphyseal plate senescence, when conditions are restored, thus enabling catch-up linear growth [9]. Clinical studies evaluating height and growth in patients with AN yielded conflicting results for stunting [10]. Growth and pubertal delay were commonly reported in 27 studies included in a meta-analysis [11]. Catch-up growth applied to the majority and a younger age and longer duration of illness were identified as potential risk factors for growth delay, whereas weight gain seemed to be associated with catch-up growth [11].

At the peak age of onset of AN at 15.5 years [4], only little further growth in stature can be expected [12]. However, in patients that develop AN before they complete puberty, weight loss or inadequate weight gain may lead to growth retardation [12]. With rising numbers of early-onset or childhood AN [13], this aspect becomes increasingly important.

The aims of the present study were:

First, to confirm the relationships of BMI, BMI-SDS and age in adolescent inpatients in a larger sample; second, to systematically assess the relationship of BMI, BMI-SDS, body height-SDS and age in adult inpatients at the time of referral; and third, to assess body height-SDS and age to evaluate stunting.

## 2. Materials and Methods

### 2.1. Sample Description

For the current analyses, admission data from patients treated between 2014 and 2021 at the Schön Klinik Roseneck, Prien am Chiemsee, Germany, a large tertiary referral center, were used. The original dataset comprised 3689 patients. We included only those 2372 female patients aged >12.0 years with a clinical diagnosis of AN according to ICD-10 who were treated at this center for the first time.

At the hospital, data from the routine diagnostic assessments (e.g., age, sex, diagnoses and body mass index) are automatically transferred to a database from which they can be exported without any identifying information (e.g., name, date of birth and place of residence) by authorized employees. Thus, accessing individual patient charts is not necessary. According to the guidelines by the ethics committee of the LMU Munich, retrospective studies conducted on already available, anonymized data are exempt from requiring ethics approval.

Body weight and height were measured using calibrated hospital scales and stadiometers. Participants were weighed in underwear without shoes. BMI values were transformed into BMI-SDSs and BMI centiles using the method suggested by [14] using nationally representative German reference data for children (KiGGS) [15] and nationally representative data for adults [16]. The calculation followed the formula SDSLMS = ([BMI/M(t)]L(t) − 1)/(L(t)S(t)), with the following abbreviations: L—Box–cox power transformation; M—median; S—variation coefficient; and BMI—individual BMI [17]. BMI-SDS approximates the deviation of an individual BMI from the median of the reference group expressed in standard deviation units. For adult patients, we used German reference data for the adult population.

Using the same method [14] and reference data [15], we also transformed individual body heights into height-SDS to investigate potential starvation-induced stunting.

Illness duration was available for 475 adult patients. In order to assess growth retardation, we included patients who developed AN before completion of puberty and the associated growth spurt (<age 16) [7,10].

### 2.2. Data Analyses

We performed the analyses in the following groups: (1) total sample: adolescents and adult inpatients, (2) adolescents only, (3) adults only.

To explore the relationship between variables we conducted Spearman correlation analyses. To compare the regression coefficients in independent groups, the z-test was used.

We used local regression (“loess”) to fit smooth curves to scatterplot data with the Epanechnikov kernel function using 50% points to fit [18]. The procedure is a generalization of least squares methods. This procedure is nonparametric in the sense that the fitting technique does not require an a priori specification of the relationship between the dependent and independent variables. Data analysis with loess allows the examination of bivariate and multivariate data, assessment of functional forms for relationships among variables, examination of model assumptions in regression analysis and representation of complex structures within data. Loess like all nonparametric fitting methods cannot be used to characterize data in terms of a simple equation [18].

We conducted curve-fitting analyses with linear, quadratic and cubic models for the association with age (z-transformed) as a predictor and BMI or BMI-SDS as an outcome.

Exact two-sided significances were calculated, and the alpha level was set to 0.005. Results with 0.005 < *p* < 0.05 were labelled as suggestive evidence [19]. All analyses were performed using IBM^®^ SPSS^®^ Statistics 27.0.0.1 and 29.0.0.0 for Windows.

## 3. Results

Descriptive data are summarized in Table 1. Mean duration of illness was more than eight years in adults and around two years in adolescents (Table 1). There was a small but significant negative correlation between illness duration and BMI in the complete sample (r = −0.12; *p* = 0.001) and in adults (r = −0.15; *p* = 0.001), but not in adolescents (r = 0.07; *p* = 0.15).

### 3.1. Relationship between Age and BMI

Average BMI at admission ranged between 14 and 15 kg/m^2^, irrespective of age (Figure 1). BMI increased with age in younger patients up to age 15, dropped slightly intermittently between ages 18 and 23 to subsequently slowly decline with age. Only adult patients had BMI values below 10 kg/m^2^ (Figure 1, Table 2 which emerged as of age 18. The number of admitted patients clearly starts dropping after age 25; this drop becomes steeper after age 30. Nevertheless, the oldest 14 patients presented at ages ≥ 60 years.

In contrast to absolute BMI, BMI-SDS was markedly lower in adolescents and to a much lesser extent in adult patients (Figure 2). In adolescents, BMI-SDS ≤ −10 occurred in only two 17-year-olds; particularly at age 18 but also thereafter, these exceedingly low BMI-SDSs occurred in small subgroups of patients up to age 51. The three lowest BMI-SDS values were observed in patients aged 35 to 41.

### 3.2. Curve Fit Analyses

The curve fit analyses for BMI with age, “age × age” and “age × age × age”, as a predictor yielded small r^2^ values for linear, quadratic and cubic equations (r^2^ = 0.006, *p* < 0.001; r^2^ = 0.007, *p* < 0.001; and r^2^ = 0.007, *p* < 0.001, respectively). For BMI-SDS, the quadratic equation fitted best according to the F-value (F (2.2369) = 143.3, *p* = 1.8 × 10^−59^ and r^2^ = 0.11) (Table 1 and Table 3, Figure 3). “Age × age” was a better predictor for BMI-SDS than age, with a break point located somewhere between ages 34 and 43 years. Interestingly, maximal BMI-SDS seemingly drops linearly with z-value for age.

Table 4 summarizes the relationship between age and BMI and BMI-SDS. Higher correlations applied to the adolescent as compared to the adult patients. There was a significant difference between groups for BMI-SDS, whereas the comparison of BMI values shows suggestive significance.

### 3.3. Association of Height, Height-SDS and BMI in Adolescents

There was a small suggestive significant correlation between BMI and height in adolescent inpatients (r = 0.09; *p* = 0.005).

### 3.4. Association of Height, Height-SDS and BMI in Adults

There was a small but significant correlation between BMI and height in adult patients, too (r = 0.1; *p* = 1.6 × 10^−4^). The mean body height-SDS was not significantly higher than 0 (0.068, *p* = 0.01, *n* = 1370) (Figure 4).

Illness duration and therefore age of onset were available for 475 adult patients. Out of these, 153 patients with a mean admission age of 24.4 years (±8.17; range 18–56) developed AN before age 16. The mean body height-SDS (M = −0.119 and SD= 0.99) in these persons did not differ significantly from 0 (t(*df* = 152) = 1.484, *p* = 0.140).

## 4. Discussion

The current study is the first to systematically assess BMI, BMI-SDS and height in relation to age in both adolescent and adult inpatients with AN at the time of referral. Mean BMI remained relatively stable across the age span with values ranging between 14 and 15 kg/m^2^. BMI initially increased with age in younger patients up to age 15, plateaued between ages 15 and 18, dropped between ages 18 and 21 and afterwards slowly declined with age. The aforementioned plateau between ages 15 and 18 [1], therefore, does not continue into adulthood; systematic BMI changes, albeit small, occur after age 18. While mean BMI values of adolescents and adults at admission were very similar, only adult patients had very low BMI values. Our findings are in line with a retrospective study of 622 female patients treated for AN on a both inpatient and outpatient basis that revealed significantly longer periods of illness and a higher rate of weight loss before the initial presentation for young adults compared to adolescents [20].

BMI-SDS was negatively correlated with age at referral in adolescents and much less pronounced in adults. The negative correlation in adolescents thus confirms the correlation obtained in the same age range in our previous study [1]. However, whereas in the former study only patients aged 15 and older contributed to the negative correlation between age and BMI-SDS, it applied to the whole age group (12 to 18 years) in the current study. The classification of the Shönklinik as a tertiary center is evidenced by an approximately one-year-longer illness duration of the adolescent inpatients of the current study as compared to those inpatients ascertained within the German Anorexia Nervosa Registry [21].

A pooled integrative data analysis (IDA) that combined data from four national population-based NIH longitudinal cohort studies showed that in a healthy sample, BMI increases from adolescence to middle age, peaks between the age of 50 to 69 years, and begins to decline after the age of 70 years [22]. Because adult patients with AN have similar absolute BMI values to those of adolescents, adults would need to achieve more weight loss, thus accounting for a lower BMI-SDS in adults as compared to adolescents. It should be noted, however, that the BMI-SDS decreased only slightly after age 18.

With a peak age of onset between ages 15 and 25 and a mean duration of illness of six years [23], many AN patients experience the transition from child mental health services to adult medicine. In Germany, patients are usually transferred at age 18. The change in treatment ethos from a “protective” approach to a “responsibility” approach [24,25] or the cessation of parental control and support, e.g., a move away from home to attend university [26], represent possible explanations for the drop in BMI in late adolescence/early adulthood and the fact that very low BMI values were only found in adults. The latter might also be attributed to a longer duration of illness. We indeed found a significant negative correlation between duration of illness and BMI in adults. In accordance with our results, a longitudinal study identified a 0.88-fold decrease in the odds of maintaining a BMI of ≥18.5 kg/m^2^ for every year increase in duration of illness [27]. Furthermore, there is growing evidence for a staging model of AN with a later stage, proposedly after seven years, in which treatment resistance leads to a shift in focus away from weight gain to improving quality of life and minimizing discomfort [28].

A retrospective study investigating the use of adult mental health services of eating disorder patients that had previously been seen in child and adolescent mental health services reported further treatment in 33% of participants, 10% of whom had extended eating disorder treatment. Patients in the extended treatment group were more likely to have a diagnosis of anorexia nervosa, were older at assessment and had a lower weight [24]. Children and adolescents might also be admitted sooner due to parental control and because they are considered more susceptible to the consequences of nutritional restriction [29].

In addition, women show a significant change in their absolute fat mass during puberty [2]. Postpubertal women might, therefore, be able to lose more weight, resulting in a lower BMI-SDS, before their health is so seriously affected that admission ensues. Patients with AN restrictive type have a lower percentage of fat mass but fat-free mass and muscle mass in kilograms are higher than in the binge/purging type—presumably due to higher levels of activity and less binge eating [30]. During refeeding, weight gain is reported to be composed of a higher increase in fat mass at the beginning, compared to fat-free mass (roughly 70% compared to 30%) [31,32]. In patients with a very low BMI (<14 kg/m^2^), the opposite seems to be the case [33]. Restoration of body fat (and weight) is crucial in the treatment of AN, considering its predictive value. Treatment success, dropout rates and prognosis seem to be associated with higher BMI values [34,35,36]. BMI is considered a reliable measure with clinical validity in children, adolescents and adults [37,38]. In adults, BMI is inversely correlated with height (r = −0.2) and has a correlation coefficient of 0.6 to 0.8 with weight [39]. In children, correlations vary according to sex and age. In prepubertal children (age 7–12), BMI shows a positive correlation with height [40]. Studies including adolescents (age > 16) show an inversion of the BMI–height relationship around puberty [41]. In contrast, our findings suggest a positive correlation between BMI and height in both adolescent and adult patients with AN.

Growth retardation with or without catch-up growth has repeatedly been reported as a consequence of restricted eating [7,11]. We were not able to detect stunting in a large sample of adolescent and adult patients. Height-SDSs of adults that developed AN before completion of puberty did not differ from zero.

The primary limitation of this study, in addition to its retrospective nature, is the inclusion of only inpatient participants, admitted to a highly specialized treatment program. Despite the high number of participants, we cannot reliably deduce that the data are representative of all patients with AN. Our analyses obviously do not cover the full spectrum of AN patients. Especially outpatients, patients with atypical AN and male patients warrant further research. According to criteria for hospital admission, we assume, for example, that inpatients present with a lower BMI than outpatients treated at the same center.

## 5. Conclusions

In conclusion, absolute BMI values seem to be relatively stable across the age span with an increase up to age 18, followed by a small decrease in early adulthood and a slow decline with age. We found that BMI-SDS is negatively correlated with age in adolescents and less pronounced in adults. There was a marked drop in BMI-SDS values during the transition period from adolescence to adulthood, making it a critical point in the treatment of AN patients. We found no evidence for stunting in adolescent and adult patients with AN.

## Figures and Tables

**Figure 1 nutrients-16-01732-f001:**
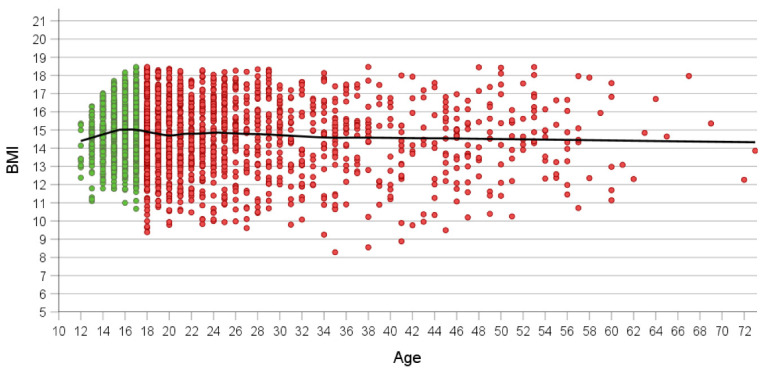
Association between age and BMI at referral in all patients (*n* = 2372). Green and red dots show adolescent (*n* = 1001) and adult patients (*n* = 1371). Loess curve is shown in black.

**Figure 2 nutrients-16-01732-f002:**
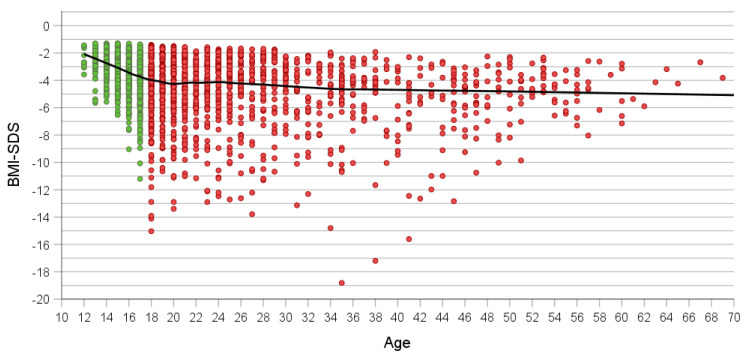
Association between age and BMI-SDS at referral in all patients (*n* = 2372). Green and red dots show adolescent (*n* = 1001) and adult patients (*n* = 1371). Loess curve is shown in black.

**Figure 3 nutrients-16-01732-f003:**
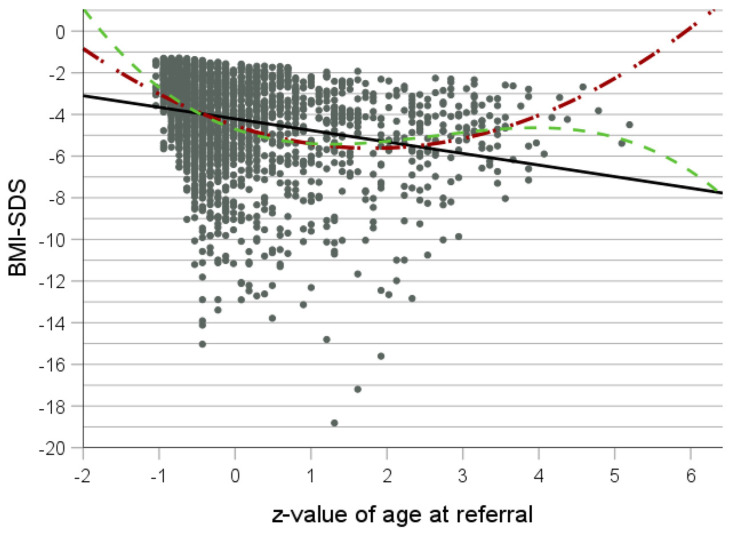
Curve fit analysis for z-value of age at referral as independent variable and BMI-SDS as dependent variable for all patients (*n* = 1472). Black line: linear equation; red line: quadratic equation; and green line: cubic equation.

**Figure 4 nutrients-16-01732-f004:**
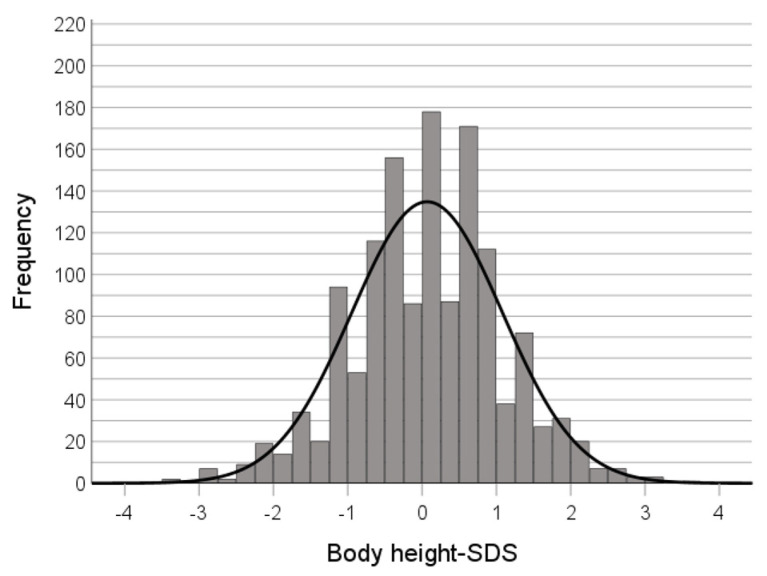
Distribution of body height-SDS at the time of admission (adult patients).

**Table 1 nutrients-16-01732-t001:** Descriptive data.

	Total	Adolescents	Adults
*n*	2372	1001	1371
AN (ICD-10: F50.0)	1454	623	831
Age (mean ± SD; range)	22.20 ± 9.78; 12–73	15.49 ± 1.26; 12–17	27.1 ± 10.36; 18–73
BMI (mean ± SD; range)	14.77 ± 1.80; 8.28–18.47	14.95 ± 1.43; 10.67–18.47	14.63 ± 2.02; 8.28–18.47
BMI-SDS	−4.21 ± 2.18; −18.81–−1.29	−3.39 ± 1.43; −11.21–−1.29	−4.81 ± 2.44; −18.80–−1.4
Duration of illness (years; mean ± SD; range) *	5.82 ± 7.49; 0–50	2.07 ± 1.26; 0–8	8.58 ± 8.70; 0–45

* data available for *n* = 381 adolescents and 475 adults.

**Table 2 nutrients-16-01732-t002:** BMI distribution in adolescents (*n* = 1001) and adults (*n* = 1371).

	*n* (%) Adolescents	*n* (%) Adults
BMI 8–9.99 kg/m^2^	0	20 (1.5)
BMI 10–10.99 kg/m^2^	2 (0.2)	42 (3.1)
BMI 11–11.99 kg/m^2^	16 (1.6)	96 (7.0)
BMI 12–12.99 kg/m^2^	71 (7.1)	123 (9.0)
BMI ≥ 13 kg/m^2^	912 (91.19)	1090 (79.5)

**Table 3 nutrients-16-01732-t003:** Results of curve fit analyses on effect of age at referral on BMI-SDS.

Dependent Variable: BMI-SDS									
Equation	Model Overview					Estimation of Parameters			
	R^2^	F	*df*1	*df*2	Sig.	Intercept	b1	b2	b3
Linear	0.06	163.3	1	2370	3.3 × 10^−36^	−4.21	−0.555		
Quadratic	0.11	143.3	2	2369	1.8 × 10^−59^	−4.54	−1.19	0.33	
Cubic	0.12	102.8	3	2368	1.5 × 10^−62^	−4.70	−1.26	0.65	0.08

Independent variable: z-score of the age at referral.

**Table 4 nutrients-16-01732-t004:** Correlations between age at referral (T0) and BMI and BMI-SDS.

	Total (*n* = 2372)	Adolescents (*n* = 1001)	Adults (*n* = 1371)	Difference between Correlation Coefficients in Adolescents and Adults: (Row Difference, z-Value, *p*-Value)
BMI kg/m^2^	r = −0.05; *p* = 0.018	r = 0.12; *p* = 2.4 × 10^−4^	r = −0.03; *p* = 0.3	0.09, z = 2.16, *p* = 0.030
BMI-SDS	r = −0.36; *p* = 9.2 × 10^−74^	r = −0.35; *p* = 2.1 × 10^−30^	r = −0.09; *p* = 0.001	0.26, z = 6.61, *p* < 0.001

## Data Availability

For questions regarding the acquisition of original data, collaboration, and other related matters, please contact the corresponding author via email for further communication due to privacy restrictions.

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
