# Peer review of "Body Mass Index Distribution in Female Child, Adolescent and Adult Inpatients with Anorexia Nervosa—A Retrospective Chart Review"

_nutrients, 2024, doi:10.3390/nu16111732_

Round 1
Reviewer 1 Report
Comments and Suggestions for Authors
In the paper, Dr. Gertraud Gradl-Dietsch et al found that BMI of AN inpatients seems to be relatively stable and BMI values initially increase with age in younger patients, drop between ages 18 and 23 and then slowly decline with age. Though this is a retrospective study and includes some bias, more than 2000 inpatients were included in this study. In order to improve their manuscript, a few suggestions were made;
1) This study included female inpatients only. Therefore, I recommend adding ‘female’ to the title.
2) Please explain the criteria for hospitalization for AN.
3) This study includes 14 patients with the age of more than 60. In general, it is hard to imagine a patient over 60 years old with AN being admitted for treatment purposes, but was the patient admitted for treatment of AN? Is there any possibility of add-on treatment for other diseases or possible age-related loss of appetite?
Reviewer 2 Report
Comments and Suggestions for Authors
Dear author, thank you for sharing your research.
The study you conducted aims to:
- confirm the relationships of BMI, BMI-SDS and age in adolescent inpatients in a larger sample,
- to systematically assess the relationship of BMI, BMI-SDS, and
body height-SDS and age in adult inpatients at the time of referral
- to assess body height-SDS and age to evaluate stunting.
Please consider to better clarify all limitations of this study.
As you rightly said, it does not involve patients who are not hospitalized or with any form of anorexia. However, the retrospective nature of the study should be emphasized, even in the title of your study.
Best regards
